Onyx disparamphis sp. n. (Nematoda, Desmodorida) from South Korea with a taxonomic review of the genus

Tchesunov Alexei V. AVTchesunov@yandex.ru 1
Jeong Raehyuk 2
Lee Wonchoel 3
1 Department of Invertebrate Zoology, Faculty of Biology, Moscow State University , Moscow , Russia
2 Department of Life Science, Chung-Ang University , Seoul , South Korea
3 Department of Life Science, Hanyang University , Seoul , South Korea
Zhukova Natalia
Electronic publication date: 2022 Mar 16
Publication date: 2022
Volume: 10
Electronic Location ID: e13010
Received 2021 Nov 5; Accepted 2022 Feb 4
Copyright: ©2022 Tchesunov et al.
Copyright year: 2022
Copyright holder: Tchesunov et al.
License: This is an open access article distributed under the terms of the Creative Commons Attribution License, which permits unrestricted use, distribution, reproduction and adaptation in any medium and for any purpose provided that it is properly attributed. For attribution, the original author(s), title, publication source (PeerJ) and either DOI or URL of the article must be cited.
License URL: https://creativecommons.org/licenses/by/4.0/

Keywords: Desmodoridae, Free-living marine nematodes, Jeju Island, Pictorial key, Taxonomy, Onyx, New species

Funding: The National Institute of Biological Sciences (NIBR) NIBR202102108 The Ministry of Environment (MOE) of the Republic of Korea This work was supported by a grant NIBR202102108 from the National Institute of Biological Sciences (NIBR) funded by the Ministry of Environment (MOE) of the Republic of Korea. The funders had no role in study design, data collection and analysis, decision to publish, or preparation of the manuscript.

==============================
A new free-living marine nematode Onyx disparamphis sp. n. (Nematoda, Desmodorida) is described from sandy littoral of Jeju Island, South Korea. The new species differs from all other Onyx species by the unusual amphideal fovea morphology in males (elongated loop). O. disparamphis relates to O. balochinensis, and O. brevispiculatum by having simple non-double terminal pharyngeal bulb and relatively small and straight, non-sigmoid supplementary organs, but differs from them by smaller body length, shorter cephalic setae, smaller terminal pharyngeal bulb, smaller spicules, number of supplementary organs and tail shape expressed as ratio tail length/anal diameter. The genus Onyx is revised with updated genus diagnosis, and an annotated list of 23 valid species is presented. Onyx ferox is considered species inquirenda because the species is known only from a sole immature female specimen, while within Onyx, the males provide the most important distinguishing characters such as enlarged and complicated amphids, supplementary organs and copulatory spicules. For species identification, a pictorial key consisting of illustrations of simplified icons of male heads and posterior body sections, as well as a table of the most important morphometric and numerical characters are provided. Geographical distribution and habitat specifity of Onyx species is analysed briefly.

Introduction

As part of the study of meiofauna and nematodes on the intertidal sandy littoral of the Jeju Island (South Korea), we have found a number of new nematode species which are partly already published (Jeong, Tchesunov & Lee, 2019; Jeong, Tchesunov & Lee, 2020; Tchesunov, Jeong & Lee, 2020; Tchesunov, Jeong & Lee, 2021). Here, we report on a new Onyx species common on this beach.

Nematode genus Onyx has been established by N.A. Cobb (1891) for a species found by him in so called Amphioxus-sand in the Bay of Naples, Italy. Cobb marked single axial spear attached to the dorsal side of the pharynx as the prominent trait of Onyx perfectus, which provided a ground to consider an evident kinship with the genus Dorylaimus. This relationship was later recognized as superficial, and Onyx was taken as a relative of Metachromadora (Filipjev, 1918: 214, at that time, Chromadoridae, Spilipherini) and then within Desmodorinae (Filipjev, 1934). Thereafter, Onyx has a stable position in the nematode system as a genus of the order Desmodorida, family Desmodoridae and subfamily Spiriniinae.

Onyx is a globally distributed, well-defined genus within the the family Desmodoridae which is mainly found in shallow coastal sediments. Species of Onyx are usually well recognizable owing to their bold and distinct structural features. Consequently, there are limited nomenclatural problems or bynonyms within the genus. However, the number of species grows, especially by the exploration of tropical meiofauna, that leads to increasing complexity in species identification. Coupled with new species description, several reviews of the genus Onyx were suggested (Blome & Riemann, 1994; Nasira, Rehmat & Shahina, 2011; Armenteros, Ruiz-Abierno & Decraemer, 2014; Huang & Wang, 2015). Increase in number of species requires periodical taxonomic revisions with proper adjustment of species composition and construction of improved keys for species identification.

Here, we propose description of a new species together with revised species list and pictorial key for species identification.

Materials & Methods

The nematodes were collected and studied in frame of a project on exploration interstitial fauna of sandy beaches of South Korea. The site of sampling is a large intertidal sandy beach Shinyang Seopjikoji at the south-eastern point of Jeju Island (Fig. 1).

Figure 1 Type locality of Onyx disparamphis sp. n.

The quantitative sediment samples were initially fixed by neutralized 5% formol on filtered sea water. Meiofauna including nematodes was separated from sediment using method of centrifugation with Ludox (Burgess, 2001), then postfixed with 70% ethanol stained with Rose Bengal. Nematode specimens were picked up under a stereomicroscope and placed in Syracuse glass with mixture of glycerine, ethanol and distilled water at a ratio of 1:29:70. After slow evaporation of ethanol and water during two days at 40° in an oven the nematodes become completely dehydrated as described by Seinhorst (1959). Nematode specimens were mounted in permanent glycerin slides within bee-wax-paraffin glass and with glass beads as separators. Specimens were studied, measured, pictured and drawn in the optical microscope Leica DM 5000 equipped with IC measure v.2.0.0.161 software and digital camera Leica DFC 425C. For scanning electron microscopy (SEM), specimens fixed in formalin in filtered sea water were then dehydrated in a graded series of ethanol-acetone solutions. Specimens were critical point-dried with carbon dioxide. Dried specimens were mounted on stubs, coated with gold-palladium mixture, and examined with a CAMScan S-2.

Type specimens are deposited in National Institute of Biological Resources (South Korea).

Nomenclatural acts

The electronic version of this article in Portable Document Format (PDF) will represent a published work according to the International Commission on Zoological Nomenclature (ICZN), and hence the new names contained in the electronic version are effectively published under that Code from the electronic edition alone. This published work and the nomenclatural acts it contains have been registered in ZooBank, the online registration system for the ICZN. The ZooBank LSIDs (Life Science Identifiers) can be resolved and the associated information viewed through any standard web browser by appending the LSID to the prefix http://zoobank.org/. The LSID for this publication is: 2FA0F335-DF23-4824-A3DC-A04DD46289BD. The online version of this work is archived and available from the following digital repositories: PeerJ, PubMed Central SCIE and CLOCKSS.

Results and Discussion

Review of the genus Onyx

Order Desmodorida Chitwood, 1936	
Family Desmodoridae Filipjev, 1922	
Subfamily Spiriniinae Gerlach & Murphy, 1965	
Genus OnyxCobb, 1891	
Diagnosis (updated after Tchesunov, 2014)	

Desmodoridae, Spiriniinae. Cylindrical body with broad rounded cephalic region and conical tail. Cuticle thin, fine but distinctly annulated, without lateral differentiation. Amphideal fovea spirally coiled in one to several turns or modified; the fovea often shifted to apical surface of the head. Buccal cavity with long spear-like dorsal tooth directed anteriorly. Terminal pharyngeal bulb mostly elongate, may be double with lens-like thickened internal cuticular lining or the lining not thickened. Numerous midventral precloacal supplementary organs tubular and in most species S-shaped. Tail conical.

Type species, Onyx perfectus Cobb, 1891. Altogether 23 valid species, all marine.

Annotated species list (names of valid species in bold)

1. Onyx adenophorus Blome & Riemann, 1994. Blome & Riemann, 1994: 1486–1488, fig. 2 A–H (males, females, juveniles); Australia, New South Wales, high-energy sandy beach.

2. Onyx balochiensis Nasira, Rehmat & Shahina, 2011. Nasira, Rehmat & Shahina, 2011: 3–4, figs 1 A-F, 2 A-E, 3 A-G, 4 A-H, Table 1 (males, females); Pakistan, Balochistan.

3. Onyx blomei Nguyen Dinh Tu et al., 2011. Nguyen Dinh Tu et al., 2011: 5–7, Table 2; figs 3, 4 (males, female); Vietnam, Ho Chi Min City, Can Gio mangrove forest, subtidal at 0.5 m depth, silt.

4. Onyx brevispiculatum (Inglis, 1963) Armenteros, Ruiz-Abierno & Decraemer, 2014. Inglis, 1963: 537–539, figs 11–15 (as Sigmophora brevispiculata) (male, females); south-west coast of South Africa, 54 m deep, mud. (Armenteros, Ruiz-Abierno & Decraemer, 2014): 24 (transfer to the genus Onyx).

5. Onyx cangioensis Nguyen Dinh Tu et al., 2011. Nguyen Dinh Tu et al., 2011: 4–5, figs 1–2 table 1 (males and females); Vietnam, Ho Chi Min City, Can Gio mangrove forest, subtidal at 0.5 m depth, silt.

6. Onyx cannoni Blome & Riemann, 1994. Blome & Riemann, 1994: 1488–1488, figs 3 A–G, 4 (males, females, juveniles); Australia, New South Wales, high-energy sandy beach.

7. Onyx cephalispiculus Hourston & Warwick, 2010. Hourston & Warwick, 2010: 56–58, fig. 6 A–C, Table 7 (males, females); south-western Australia, subtidal sediment of heterogeneous grain size with low to moderate particulate organic content.

8. Onyx cobbi Nguyen Dinh Tu et al., 2011. Nguyen Dinh Tu et al., 2011: 9–13, figs 7–8, Table 4 (males, females); Vietnam, Ho Chi Min City, Can Gio mangrove forest, subtidal at 0.5 m depth, silt.

9. Onyx dimorphus Gerlach, 1963. Gerlach, 1963: 74, Fig. a–f, Taf. 4 (male, female); Maldives, Fadiffolu-Atoll, coarse sand.

10. Onyx disparamphis sp. n. Present paper.

11. Onyx ferox (Ditlevsen, 1921) Gerlach, 1951. Ditlevsen, 1921: 4–6, Fig. 3 and pl. 1 Figs 2, 10, 11 (as Oistolaimus ferox) (single female); Subantarctic Pacific, Auckland Islands, Carnley harbour, clay. (Gerlach, 1951): 61 (transfer to Onyx). Since the species is up to date known by an only female specimen not fully sexually developed, Onyx ferox is considered here species inquirenda.

12. Onyx litorale (Schulz, 1938) Armenteros, Ruiz-Abierno & Decraemer, 2014. Schulz, 1938: 119–121, Abb. 2 Fig. 12, 13, 14, 15 (as Parachromadora littoralis) (males, females); Amrum Island, North Sea, sandy intertidal zone. Gerlach, 1951: 61 (transfer to Sigmophora). Gerlach, 1951: 73–74, Abb. 7 a–d (as Sigmophora litoralis) (male, female); Amrum Island, North Sea, sandy intertidal zone (despite the description text refers to fig. 7, the legend of the fig. 7 indicates Sigmophora rufum and fig. 6 indicates Sigmophora litoralis; we consider the legend of fig. 6 right since that image corresponds to the text description). Armenteros, Ruiz-Abierno & Decraemer, 2014: 24 (transfer to Onyx).

13. Onyx macramphis Blome & Riemann, 1994. Blome & Riemann, 1994: 1484–1486, Fig. 1 A–G (males, females, juveniles); Australia, New South Wales, high energy sandy beach.

14. Onyx mangrovi Nguyen Dinh Tu et al., 2011. Nguyen Dinh Tu et al., 2011: 17–19, Table 6, Fig. 11, 12 (males, females); Vietnam, Ho Chi Min City, Can Gio mangrove forest, subtidal at 0.5 m depth, silt.

15. Onyx minor Huang & Wang, 2015. Huang & Wang, 2015: 1129–1130, Figs 3, 4 (males, females); Yellow Sea, intertidal sandy sediment.

16. Onyx monstrosum (Gerlach, 1956) Armenteros, Ruiz-Abierno & Decraemer, 2014. Gerlach, 1956: 431–433, fig. 4 a–g (as Sigmophora monstrosum) (males); Bay of Biscay, coastal ground water. Armenteros, Ruiz-Abierno & Decraemer, 2014: 24 (transfer to the genus Onyx).

17. Onyx orientalis Nguyen Dinh Tu et al., 2011. Nguyen Dinh Tu et al., 2011: 8–9, Table 3, Fig. 9, 10 (males); Vietnam, Ho Chi Min City, Can Gio mangrove forest, subtidal at 0.5 m depth; Quang Ninh province.

18. Onyx paradimorphus Nguyen Dinh Tu et al., 2011. Nguyen Dinh Tu et al., 2011: 13–16, Table 5, Fig. 5, 6 (males, females); Vietnam, Ho Chi Min City, Can Gio mangrove forest, subtidal at 0.5 m depth.

19. Onyx perfectus Cobb, 1891. Cobb, 1891: 153–155, figs 4–5, 7–8 (male, female); Mediterranean, Bay of Naples, sand with Amphioxus. Filipjev, 1918: 214–218, Fig. 41 a–e (males, females); Black Sea, sand with Amphioxus. Gerlach, 1963: 73–74, Taf. 3, k–l (male) (as Onyx cf. perfectus); Maldive Islands, 10 m deep, sand. Riemann, 1966: 149–150, Abb. 38 a–h (males, female); North Sea, 9–27 m deep, fine to medium sand. Specimen designated as Onyx cf. perfectus by Gerlach (1963) (Maldive Islands) differs significantly from the original description and redescriptions by much lesser body length (688 µm), a (16), cephalic setae length (8.5 µm) and lesser number of supplementary organs (11), and hence could not be considered as O. perfectus.

20. Onyx potteri Hourston & Warwick, 2010. Hourston & Warwick, 2010: 58–60, figs 7 A–D, Table 8 (male, female, juveniles); western Australia, calcareous sediment at relatively high-energy site.

21. Onyx rizhaoensis Huang & Wang, 2015. Huang & Wang, 2015: 1128–1129, Figs 1, 2 (males, females); Yellow Sea, intertidal sandy sediment.

22. Onyx rugatus Wieser, 1959. Wieser, 1959: 47–48, Fig. 48 a–d (males, females); Pacific coast, of USA, Puget Sound, sandy beach.

23. Onyx sagittarius Gerlach, 1950. Gerlach, 1950: 190–193, Abb. 5 a–e (male, females, juvenile); North Sea, sand. Gerlach, 1953: 562–563 (male, female); Mediterranean, Tyrrhenian Sea, coastal ground water.

24. Onyx septempapillatus Wieser, 1954. Wieser, 1954: 51–52, Fig. 125 a–d (male, females); Chile, littoral, exposed sand.

Description of new species of Onyx

Onyx disparamphis Tchesunov, Jeong & Lee sp. n.	
Figs. 2–6, Table 1	
urn:lsid:zoobank.org:act:C403A850-22EB-43A2-9A51-E2CCCB36364A	

Etymology

The species name reflects strong sexual dimorphism in amphideal fovea outline.

Figure 2 Onyx disparamphis sp. n., entire: Male (holotype) (A); Female (paratype) (B).

Scale bars 100 µm.

Figure 3 Onyx disparamphis sp. n., anterior ends. Male holotype, surface view (A). Male holotype, optical section (sensilla not depicted) (B). Female paratype (C).

Scale bars 20 µm. Arrow indicates a circular pore.

Figure 4 Onyx disparamphis sp. n., details of the male holotype: Anterior body (A); Posterior body (B).

Scale bars 20 µm.

Figure 5 Onyx disparamphis, optical micrographs: Male paratype, entire (A). Head of a juvenile specimen, optical section (B). Male paratype tail (C). Male paratype copulatory apparatus and posterior supplementary organs (D).

Scale bars: A –100 µm; B–D –1.

Figure 6 Onyx disparamphis sp. n., details in SEM-micrographs: (A) male head, long amphideal fovea; (B) male head, subapical view; (C) female head, subapical view; (D) male posterior body; (E) male supplementary organs (anterior body end to the left).

Scale bars: A, B: 10 µm; C, E: 3 µm; D: 30 µm.

Material examined

Holotype male, 17 paratype males and 13 paratype females are deposited in the National Institute of Biological Resources (South Korea). Inventory numbers of the holotype male is U1-5 r2 sl8, paratype males 1 and 2 are M1-5 r1 sl12, paratype male 3 and paratype female 1 –slide M1-5 r1 sl14, paratype females 2 and 3 –slide M1-5 r2 sl6, paratype males 4, 5 and 6 –slide U0-1 r2 sl15, paratype females 4 and 5 –slide U15-20 r1 sl3.

Type locality

Intertidal zone at coast of Jeju Island, South Korea (33°26′05″N, 126°55′15″E), sandy beach, June 19, 2019.

Description

Males. Body cylindrical, anterior end rounded truncated, tail conical (Figs. 1A, 4A). Cuticle thin, fine but distinctly cross annulated, without a lateral differentiation. Numbers of cuticular annules within 10 µm varies along the body: 14 annules within 10 µm at the level of the long amphideal branch and 18 less distinct annules within 10 µm at the midbody.

Apical region of the cephalic region not annulated but finely longitudinally striated, with sharp border between longitudinal striation and annulation (Figs. 6B, 6C). Inner labial sensilla not evident. Outer labial sensilla as six minute papillae (1–1.5 µm). Four long cephalic setae situated apically and directed anteriorly. There first, anterior crown of eight subcephalic setae located just posterior to the cephalic setae at the level of the anterior margin of cross annulation of the cuticle. The second, posterior, less regular crown of eight subcephalic setae located slightly anterior to the middle point of the long amphideal branch. The subcephalic setae of both crowns and several irregular singular setae in the preneural body region are about equal in length and breadth to one another and to cephalic setae. Other shorter somatic setae are dispersed sparsely along the body. There are cuticular pores distributed along the body; which look like a minute hole in the centre of a smooth circular spot on the cuticle (Fig. 3A, arrow).

Amphids shifted onto the apical area of the cephalic region. Anterior end of the amphideal fovea is located on apical surface close to the mouth opening; the fovea turns dorsally and runs on into the long dorsal arm of the fovea extended far hindward as elongated loop with tight but distinct ridge-like interspace between two arms (Figs. 3A, 3C; 6A–6C).

Somatic cuticle not widened around the mouth. Cheilostoma shaped as a truncate cone with longitudinal rugosity. Long and narrow pharyngostoma armed with a long dorsal tooth provided with a conical pointed cuticular arrowhead (corona). The tooth is adherent dorsally to the pharynx tissue at two thirds of its length. Pharyngostoma is surrounded by inflated pharyngeal tissue with fine transversal striation. Middle part of the pharynx slender; posterior part of the pharynx is formed as an elongate terminal bulb with muscular cross striation. Internal cuticular lining of the bulb with muscular cross striation not thickened and seemingly not modified. Midgut slender, filled with orange pigment inclusions.

No renette found.

Table 1 Morphometrics of Onyx disparamphis sp. n. type specimens.

Character	Holotype male	Males (holotype and paratypes together)	Female paratypes	
		n	min–max	mean	SD	CV	n	min–max	mean	SD	CV	
Body length, µm	840	18	7503–1014	873	60.5	6.94	13	695–889	800	55.1	6.89	
a	33.1	17	28.8–40.5	34.0	3.26	9.58	13	20.0–27.8	22.5	2.15	9.54	
b	5.46	17	4.66–6.26	5.55	0.37	6.67	13	4.59–5.40	4.98	0.26	5.22	
c	13.8	17	9.35–13.8	11.2	1.26	11.3	12	9.71–14.2	11.4	1.16	10.2	
c’	3.54	13	2.48–4.34	3.47	0.54	15.6	12	2.48–4.11	3.36	0.39	11.6	
V, %	–	–	–	–	–	–	12	46.4–53.5	50.6	2.24	4.43	
Body diameter at the level of the subcephalic setae, µm	24	12	20.0—25.9	23.9	1.83	7.66	10	22.0–34.0	27.7	3.67	13.3	
Body diameter at the level of the nerve ring, µm	26	18	23.4–28.9	26.0	1.31	5.04	13	27.6–31.4	29.4	1.17	3.98	
Body diameter at the level of the cardia, µm	26	18	24.1–30.0	26.1	1.37	5.24	13	29.0–37.0	32.5	2.38	7.33	
Body diameter at the level of the midbody, µm	25	17	23.9–28.0	25.7	1.05	4.09	13	32.0–40.0	35.6	1.85	5.19	
Body diameter at the level of the cloaca/anus, µm	23	13	21.0–25.6	23.3	1.33	5.72	13	19.0–25.0	21.4	1.59	7.45	
Cephalic setae length, µm	12	16	9.30–14.2	11.4	1.22	10.7	11	6.50–13.0	10.2	1.98	19.3	
Subcephalic setae length, µm	13	12	8.40–13.4	11.8	1.37	11.6	6	9.60-12.5	10.5	1.10	10.5	
Amphid width anteriorly, µm	6	5	5.30–7.50	6.54	0.88	13.5	5	6.80–8.00	7.34	0.61	8.31	
Amphid furrow length, µm	73	13	340–100	73.0	16.6	22.8	–	–	–	–	–	
Stoma total length, µm	53	16	42.0–66.0	49.8	6.52	13.1	12	43.6–62.0	53.0	5.51	10.4	
Dorsal tooth length ventrally, µm	9	14	26.8–39.6	34.4	3.48	10.1	11	33.0–40.3	29.4	12.2	41.7	
Dorsal tooth length dorsally, µm	40	14	6.80–9.80	8.30	0.97	11.7	12	5.00–19.9	9.67	3.96	41.0	
Terminal bulb length, µm	41	17	38.6–50.0	44.5	3.19	7.17	13	49.2–62.0	55.4	4.15	7.50	
Terminal bulb width, µm	18	18	15.5–19.9	17.1	1.12	6.53	13	17.8–27.0	22.9	2.42	10.6	
Number of precloacal supplements	18	13	14–19	16.9	1.52	9.02	–	–	–	–	–	
Spicules, length along the arc, µm	35	16	33.2–40.0	36.3	1.91	5.26	–	–	–	–	–	
Spicules, length along the chord, µm	26	16	23.4–30.4	26.7	1.99	7.45	–	–	–	–	–	
Gubernaculum length, µm	16	16	12.3–19.5	15.5	1.97	12.7	–	–	–	–	–	
Notes.

n, number of individuals; min–max, range, SD, standard deviation; CV, coefficient of variation (SD divided by mean, in %%)

Testis singular anterior, outstretched, situated to the left of the intestine in all the male specimens. Spicules paired and equal, short, arcuate, proximally cephalated and distally pointed. Gubernaculum S-shaped and oriented perpendicularly to the longitudinal body axis. Series of 14–19 equal midventral precloacal supplementary organs. Supplements consist of three constituents, (1) surface cuticular pit with cuticularized walls, (2) core within the pit, head of the core bears a longitudinal ridge with a papilla, (3) short internal straight cuticular tube extending from the pit obliquely inward (Figs. 4B, 6D–6E).

Tail conical, with caudal glands and a terminal spinneret. Caudal gland cell bodies hardly discernible, visibility limited within tail, but may have seemingly pushed out to preanal region in some specimens.

Females. Amphideal fovea spirally coiled in three turns and situated entirely on the cephalic apex close to the mouth opening (Figs. 3C, 6C).

Ovaries paired, antidromously reflected, both situated to the left of the intestine in all female specimens studied (Fig. 1B).

Table 2 Characters of Onyx species (males).

Species	Characters	
	Body length	a	Cephalic setae length	Dorsal tooth length	Terminal pharyngea l bulb, length & shape, internal lining	Number of supplements	Spicule length	c’	
adenophorus	1154	41	10	23	50, elongate, double, lining cuticularized	18, sigmoid	29	2.1	
balochiensis	1240–1640	39–52	17–19	32–36	84–100, inconspicuously double, lining faint	15–23, tubular	45–50	2.3–3.2	
blomei	697–756	25–28	6–6.5	26–29	50, elongate, lining faint	7–8 tubular, slightly S-shaped	34–35	3.5–3.7	
brevispiculatum	1800	22	?	?	no true bulb but slight swelling, lining faint	39 crochet-shaped	117	1.2 calc	
cangioensis	679–759	20–23	5	31–36	29 calc, oval, lining faint	14–16, tubular	45–49	2.4–3	
cannoni	1032	44	8	22	46 calc, elongate; lining faint	15, tubular	28	∼2 calc	
cephalispiculus	1204–1284	20–28	12–15	∼40 calc	70 calc, elongate, double, lining cuticularized	18–24, S-shaped	65–75	2.4 calc	
cobbi	1334–1544	31–40	21–22	39–43	80 calc, elongate, double, lining cuticularized	15–16, tubular	42-46	3–3.7	
dimorphus	1080	21	15–20	50	66–85, elongate, lining faint	10, S-shaped	45	2.6	
disparamphis	750–1014	28–41	9.3–15	26–40	38–50, elongate, lining faint	14–19, straight tubular	33–40	2.5–4.4	
litorale	1100–1360	28 & 20	15	35	64, elongate, lining faint	15–20, sigmoid	190–200	3.4 calc	
macramphis	813	37	8	29	42, elongate, lining faint	14, tubular	27	2.9	
mangrovi	523–591	12–15	3 & 3	30–36	70 calc, lining faint	17–23, tubular sigmoid	36–40	1,1–1,4	
minor	675–806	35–41	7	21–22	33–40, elongate, double, lining lens-like cuticularized	12, tubular S-shaped	22–25	3.3–3.5	
monstrosum	1865–2059	43–49	13	?	81, double, lining lens-like cuticularized	19–21, sigmoid	55–60	3	
orientalis	974–1003	39–44	15–17	24–27	76 calc, double, lining lens-like cuticularized	17–18, tubular	38–40	2.5–2.8	
paradimorphus	1003–1196	25-31	18	44	70 calc, double, lining lens-like cuticularized	15, sigmoid tubular	41–44	4.2–4.3	
perfectus	1740–2160	37–39	22–28	53–56	88 calc, elongate, lining faint	13–22, sigmoid	45–70	2	
potteri	1112	40	14	?	55 calc, elongate, internal lining faint	10, slightly sigmoid	50	3 calc	
rizhaoensis	1213–1330	44–45	10	20–22	55 calc, elongate, double, lining lens-like sclerotized	12, sigmoid	30	2.6–2.8	
rugatus	1300	32–33	19	40 calc	90, double, lining lens-like cuticularized	22, complicated papillae	42	2.7	
sagittarius	1070	28	5	30	60, double, lining faint	24, slightly sigmoid	35	2	
septempapillatus	1320	40	20	21	75, double, lining cuticularized	7, small and nearly straight	37	3	

Diagnosis

Body length 695–1014 µm, index a 20–40.5, index c’ 2.5–4.34. Cephalic setae 6.5–14.2 µm long. Two subsequent crowns of eight subcephalic setae similar in length to the cephalic ones. Amphideal fovea shows strong sexual dimorphism: in females, the amphideal fovea spirally coiled in three turns and located entirely on the head apex, while in males, the fovea turns dorsally from the aperture on the apex, then extended hindward as elongated loop. Dorsal tooth 27–40 µm along the ventral side. Terminal bulb of the pharynx elongated, 40–60 µm long, with faint internal lining. Spicules arcuate, 33–40 µm long. Midventral precloacal supplementary organs 14–19 in number, consist of flat cap and internal short and almost straight cuticular tubes. Tail conical, c’ 2.4–4.3.

Relationships

Onyx disparamphis sp. n. differs immediately from all other Onyx species by the peculiarly very long loop of the amphideal fovea of males. O. disparamphis shares simple non-double terminal pharyngeal bulb with thirteen other Onyx species (Table 2), among them O. balochinensis, and O. brevispiculatum may show some resemblance in shape of supplementary organs, relatively small and straight, non-sigmoid. O. disparamphis differs from O. balochinensis by smaller body length (690–1014 versus 1240–1640 µm), smaller cephalic setae (6.5–15 versus 17–22 µm), smaller terminal pharyngeal bulb (38–62 versus 84–105 µm in length) and smaller spicules (33–40 versus 45–50 µm). O. disparamphis differs from O. brevispiculatum by also smaller body (695–1014 versus 1800–3300 µm), twice the lower number of supplementary organs (14–19 versus 39) and relatively longer tail (c’ 2.4–4.4 versus 1.2).

Ecological remarks

Onyx disparamphis is a common, but not very numerous species on the Shinyang beach, being the 12th most abundant and comprising 1.6% of total nematode abundance of the beach. O. disparamphis is distributed across the whole intertidal sandy beach from lower to upper horizon, with some increase in numbers at middle and upper horizons. No obvious confinedness to a certain layer in vertical sediment column is observed, evidently because of uniformity of conditions (granulometry) within a sediment depth 0–20 cm.

Figure 7 Pictorial key to Onyx species, male heads. Beginning.

Figure 8 Pictorial key to Onyx species, male heads. Continuation.

Figure 9 Geographical distribution of Onyx species.

Species are designated with three first letters of the names. Type localities are in bold.

Microscopic examination of gut content does not reveal any evident particles or identifiable remnants. The intestine of the individuals studied either appear empty or with spherical drops. The long and strong dorsal tooth may protrude from the mouth as it is shown for Onyx sagittarius by Gerlach (1950, Abb. 5c). We suppose that Onyx disparamphis and its congeners could pierce covers of some food items (e.g., protists) and suck out the liquid matter by muscular bulb pumping.

Pictorial key for identification Onyx species

The present pictorial key for identification of 23 valid species of Onyx is constructed based on principles of Platt (1984), who first introduced such keys in marine nematology. The key consists of two components, (1) a set of species icons or pictorial key (Figs. 7–8), and (2) a table of the most important morphometric and numerical characteristics (Table 2). Most valid species are known on both males and females, and only two species, O. monstrosum and O. orientalis are described based only on males. Only males are used for pictures since they provide more distinctly perceiving features (such as amphids) on the head and much more features (such as copulatory and supplementary organs) on the posterior body. The table 2 includes characteristics of only males, because females are not known for two species and data on females are often less complete in comparison with males in other descriptions based on two sexes.

On Figs. 7–8, the heads are arranged in such an order that species having the largest and most conspicuous amphideal fovea are located at the top of the key; by scanning the icons from top to bottom, the amphids and head setae become gradually smaller and shorter.

Spatial distribution of Onyx species

Like most other marine nematode superspecies taxa, the genus Onyx shows a worldwide distribution (Fig. 9); however, most species are confined with warm waters, and none occurs in Arctic and Antarctic (with possible exception O. ferox sp. inq. in Subantarctic). Nine species are recorded in tropical areas (balochiensis, blomei, cangioensis, cobbi, dimorphus, mangrove, orientalis, paradimorphus, perfectus); eight species in subtropical regions (adenophorus, brevispiculatum, cannoni, cephalispiculus, macramphis, perfectus, potteri, sagittarius), nine species in temperate regions (disparamphis, littorale, minor, monstrosum, perfectus, rizhaoensis, rugatus, sagittarius, septempapillatus). Surprisingly, there were no Onyx species recorded along the east coasts of the Americas, but this is likely due to insufficient information on meiofauna in this specific region. A cluster of six co-occuring species within a limited area was found in the CanGio mangrove habitat (South Vietnam) in silty sediment at a depth 0.5 m.

All the Onyx species are confined with coastal shallows, from intertidal zone to upper sublittoral of several tens of meters depth; no species are recorded from the deep sea. The majority of species inhabits coarse sands (16 of 24 species), often on high energy beaches. The remaining eight species were found on silts, most of them (six) in mangrove milieu.

Conclusions

The sandy intertidal at Shinyang, Jeju Island, is distinguished by its high nematode diversity. We have recorded over 90 nematode species belonging to 73 genera, 31 families and eight orders. To date, only ten species are identified up to species level and five of them having been proved new for science are published. The next taxa to be treated are the most diverse families Xyalidae and then Chromadoridae composing respectively 17% and 14% of all species revealed there.

Supplemental Information

Supplemental Information 1 Raw morphometric data of Onyx disparamphis sp. n

The table of raw data contains all morphometric data for all type specimens of Onyx disparamphis sp. n. The data allow distinguish Onyx disparamphis fron all other Onyx species.

Left column presents dimensions, measurement and characters which are used normally for species discriminations. Uppermost line presents type specimens in slides.

Males and females are given on different sheets. Meanings of characters are not abbreviated and presented as is in the left column.

Statistical parameters in the right columns: n –number of specimens measured; min –minimal variable; max –maximal variable, mean –mean value, SD –standard deviation, CV –coefficient of variation (standard deviation divided by mean value, in %%).

Some variables are missing because of impossibility of measuring (damaged specimen, inconvenient location in slide) and replaced by “x”.

Click here for additional data file.

The authors thank Dr. Vadim Mokievsky (P.P. Shirsov Institute of Oceanology, Moscow) and Mrs. Maria Fedyaeva (Moscow State University) for their help in the field work sampling in June 2019. We appreciate the reviewers for their valuable remarks.

Abbreviations

a body length divided by body diameter at midbody

b body length divided by pharynx length

c body length divided by tail length

c’ tail length divided by anal body diameter

calc calculated or measured from published measurements and/or figures

V distance from anterior end to vulva divided by entire body length, in %%

Additional Information and Declarations

Competing Interests

Author Contributions

Data Deposition

New Species Registration

The authors declare that there are no competing interests.

Alexei V. Tchesunov, Raehyuk Jeong and Wonchoel Lee conceived and designed the experiments, performed the experiments, analyzed the data, prepared figures and/or tables, authored or reviewed drafts of the paper, and approved the final draft.

The following information was supplied regarding data availability:

The raw data are available as a Supplemental File.

The following information was supplied regarding the registration of a newly described species:

Publication LSID: urn:lsid:zoobank.org:pub:A76B35DE-E24C-4EA9-9CB2-8DE1A95D10AB

Onyx disparamphis LSID: urn:lsid:zoobank.org:act:FAF768CE-5F30-40F2-896A-2670FA2AE35E.

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
