# Peer review of "Onyx disparamphis sp. n. (Nematoda, Desmodorida) from South Korea with a taxonomic review of the genus"

_PeerJ, doi:10.7717/peerj.13010_

## Round 0.1 · original submission · Minor Revisions

Dear Dr. Tchesunov,

We are pleased to let you know that your manuscript has now passed through the review stage and is ready for revision. The reviewers provided detailed comments, and we ask that you consider these carefully and correct the errors when revising the manuscript.

To ensure the Editor and Reviewers will be able to recommend that your revised manuscript is accepted, please pay careful attention to each of the comments. Please submit a point-by-point response to the comments and a rebuttal of any criticisms or requested revisions that you disagreed with.

I look forward to receiving your revision.

Natalia Zhukova

Reviewer 1 ·

Basic reporting

No comments here.

Experimental design

No comments here.

Validity of the findings

No comments here.

Additional comments

Line 24: "relatively tail shape" is grammatically incorrect,, something is missing.
Line 30: "posterior bodies" replace with "posterior body sections".
Line 51: "quite few confusions" makes no sense, "few" means "not many" while "quite a few" means "many" so please clarify what is meant here.
Line 52: "growths" replace with "grows"
Line 66: "drawn out" replace with "extracted".
Line 109 and below: Please provide original binomen (basyonym) for all species that were not originally described in Onyx.
Lines 252-255: Description of the female reproductive system is brief and without any details. Please describe every element of female gonads in details. Please also provide detailed close-up drawings of female gonads.
Lines 308-309: Exactly, so please do not follow the same trend and expand the description of a female sexual characters in your descriptions.
Lines 310-317: In my opinion, it makes more sense and it is much easier to use when heads and tails of each species are paired up in the illustrations.
Figure 11: Please give full names for all species in the figure legend.

Reviewer 2 ·

Basic reporting

Manuscript is devoted to the description of a new species of free living marine nematode belonging to the family Desmodoridae. In present time ex[anding of our knowledge in marine biodiversity is one of the important and actual field of scientific investigations. So, manuscript must be published in PeerJ journal.
I am not a native English speaker and cannot evaluate this aspect of manuscript.
Manuscript is written in professional language with appropriate structure, qualitative figures and tables.
However I have some remarks.
General remark - check all the figures. Number of figures in the text does not correspond with figures itself.
Line 96. Check the author of subfamily. Should be Chitwood, 1936
Line 113. Must be Shahina instead of "Shabina"
Line 210-212 "There are 14 distinct annules within 10 μm at the level of the middle of the long amphideal branch and about 18 less distinctly discernible annules within 10 μm at the midbody" What is this sentence about? Please, emplane and show in figures.
Line 219. "The second, posterior, less irregular crown of eight subcephalic setae located slightly posterior to the middle point of the long amphideal branch". I do not agree with this. There is second circle of subcefalic setae not far from the first one (fig 3BC, 6B), whereas another circle of setae in the middle of amphid are also there, I would call them cervical setae.
Line 223. Please, show this pores either on light microscopy or SEM photos.
Line 243. There are instead of "there a"
Line 311. Must be amphidial fovea instead of "amphids fovea"
Table 1. Check the body length, is it really 7503-1014?
Figure 4A - where are setae?
Figure 5. What is scale bar for C-D? Is it 1 mkm?
Figure 6. What the scale?

Experimental design

no comments

Validity of the findings

no comments

Reviewer 3 ·

Basic reporting

see attached document

Experimental design

see attached document

Validity of the findings

see attached document

Additional comments

see attached document

Annotated reviews are not available for download in order to protect the identity of reviewers who chose to remain anonymous.

---

## Round 0.2 · accepted · Accept

Dear Dr. Tchesunov,

Thank you for submitting the revised version of your manuscript to PeerJ.

In the revised version the authors took into consideration all comments and remarks. I recommend accepting your manuscript for publication in PeerJ.

Natalia V. Zhukova